# Effect of FSP on Tribological Properties of Grade B89 Tin Babbitt

**DOI:** 10.3390/ma14102627

**Published:** 2021-05-17

**Authors:** Marcin Madej, Beata Leszczyńska-Madej, Joanna Hrabia-Wiśnios, Aleksandra Węglowska

**Affiliations:** 1Faculty of Metals Engineering and Industrial Computer Science, AGH University of Science and Technology, 30 Mickiewicza Ave., 30-059 Krakow, Poland; 2Faculty of Non-Ferrous Metals, AGH University of Science and Technology, 30 Mickiewicza Ave., 30-059 Krakow, Poland; bleszcz@agh.edu.pl (B.L.-M.); hrabia@agh.edu.pl (J.H.-W.); 3Lukasiewicz Research Network—Welding Institute, 16-18 Błogosławionego Czesława Ave., 44-100 Gliwice, Poland; Aleksandra.Weglowska@is.gliwice.pl

**Keywords:** bearing alloys, FSP, tribological properties, wear mechanism

## Abstract

The article presents the results of tests of the tribological properties of a bearing alloy on a tin matrix (B89) after surface modification by means of friction stir processing (FSP) with a Whorl tool. The samples were processed using two tool speeds of 280 and 580 RPM and a constant linear speed of 355 mm/min. The obtained results proved the significant influence of FSP on both the morphology of the precipitates and the investigated properties. Changes in the nature and course of friction were also observed, including the participation of individual wear mechanisms in the studied test conditions. The use of the 560 RPM tool rotation speed reduces the friction coefficient and the weight loss by about 30%. The flexural strength was also increased from 123 to 307 MPa.

## 1. Introduction

Appropriate sliding properties (including wear and galling resistance), strength and anti-corrosion properties determine the suitability of an alloy as a material for sliding bearing bushes. The combination of such diverse properties can only be achieved by alloys composed of two or more phases with different properties. The basic factor determining the application of Sn-Sb-Cu alloys for bearing alloys is the presence of hard SnSb compound crystals and acicular particles of the Cu_6_Sn_5_ compound, evenly distributed in a soft, plastic tin-rich matrix [1,2,3,4,5]. Previous studies show that refinement and a change in the morphology of the hard SnSb intermetallic phases improve, in particular, the tribological properties. Damage to sliding bearings is determined by the coexistence of two mechanisms: mechanical fatigue due to the cyclical nature of the load on sliding bearings, as well as friction wear due to the contact of the Babbitt bushing/steel shaft friction pair and the presence of solid particles in the lubricant (wear products, admixtures) acting as an additional abrasive. Barykin et al. [2] showed the influence of β phase (SnSb) particle size refinement on the change in bearing alloy wear intensity. The results of the tests carried out under conditions of technically dry friction presented by the authors proved that that the use of heat treatment reduces the wear intensity of the tested bearing alloy by 25%, and after the application of the plasma spray obtained by as much as 40% compared to the alloy in the cast condition. Potekhin et al. [3] studied various casting methods, including the turbulent casting method, making it possible to obtain a tin bearing alloy with a microstructure with globular precipitates of intermetallic phases, which in turn leads to the most favorable tribological properties. On the other hand, the lubricated forging method proposed by the authors of work [4] contributed to the production of a homogeneous microstructure with numerous fine and equiaxed SnSb phase particles with a size of 1–2 μm, which resulted in a significant increase in the share of ductile cracking, the main feature of which is the slow development of cracks. Moreover, the authors proved that the rate of tin Babbitt wear depends on the wear rate of β phase particles (SnSb). Similarly, works [6,7,8,9] present the influence of coating application methods on the microstructure and properties of Babbits. Additionally, the positive effect of changing the morphology of SnSb precipitates on reducing the tribological wear was also presented in [2,10,11,12]. Large, angular precipitates of the SnSb phase with sharp edges easily detached from the matrix, leaving voids behind and creating deep scratches during bearing operation, at the same time accelerating its wear. The heat treatment proposed by B. Leszczyńska-Madej [11,12] enables the refinement of and a change in the morphology of precipitates occurring in the alloy, lowering the coefficient of friction of the friction pair, thus reducing the wear rate under lubricated working conditions [11]. On the other hand, the authors of work [13], as a result of thermo-plastic treatment, obtained an anti-friction layer with a fine-grained microstructure and evenly distributed β phase precipitates (SnSb), which also contributed to a reduction in the wear intensity. According to the research presented in [14], high plastic deformation in the process of equiangular channel pressing (ECAP) leading to a reduction in grain size and strong refinement of the intermetallic phase particles (SnSb, Cu_3_Sn) causes even a fourfold reduction in the wear rate and a decrease in the coefficient of friction of the Babbitt–steel pair. This is due to the formation of developed porosity on the deformed surface of the bearing alloy, which improves the lubrication conditions of the Babbitt–steel friction pair. At the same time, unfortunately, under dry friction conditions an increase in the wear rate of the tin alloy was noted, mainly owing to numerous defects (microcracks) that are introduced into the structure of the tin alloy after its strong plastic deformation, and thus reduce the fatigue wear resistance of the surface layer of the studied material. 

FSP is a new but very promising surface modification method that is only just entering the technological application phase. In a relatively short time since its invention [15], FSP technology has found many applications, both for modifying and improving the properties of foundry products [16,17,18,19], sinters obtained by powder metallurgy technology [20], as well as for introducing additional materials into the surface layers of a processed material (friction stir alloying—FSA) [21,22]. Dynamic recrystallization, which takes place in the process zone during modification with the FSP method [23,24], leads to microstructure refinement, increases fatigue strength [19] and improves mechanical [16,17] and tribological [25,26] properties.

The aim of the study was to determine the effect of FSP using the Whorl tool, on the microstructure and tribological properties of the B89 bearing alloy. The conducted tribological tests allowed the authors to determine how the change in the morphology of the precipitates present in the alloy affects the wear resistance and the course of friction. Earlier studies by the authors on the B83 alloy [26] modified by FSP with the use of the Triflute tool proved the improvement of the tribological properties as a result of refinement of the hard phases present in the alloy after FSP modification. This article focuses on the characteristics of the tribological properties of the B89 alloy after FSP modification using the Whorl tool with a different geometry. The research is innovative as the authors were the first to use the FSP method to modify the surface of bearing alloys.

## 2. Materials and Methods

The material for the research was a tin-based casting alloy: SnSb9Cu4 (B89) used for casting plain bearing bushings, the chemical composition of which is presented in Table 1. The chemical composition was determined using a scanning microscope based on analysis of the sample area.

The microstructure and tribological properties of the alloy were investigated in the initial (cast) state and after modification by FSP. The FSP surface modification was performed on a FYF32JU2 (Jafo, Jarocin, Poland) vertical milling machine welding stand. The process was carried out using a Whorl-type pin dedicated for soft materials-cone-shaped with a spiral-shaped thread (Figure 1) made of high-speed steel HS6-5-2, with two different rotational speeds of the tool-280 and 560 RPM-and the constant linear speed of 355 mm/min.

Analysis of the microstructure of the studied materials along with analysis of the chemical composition in micro-areas was performed using a scanning electron microscope (Hitachi SU 70, Tokyo, Japan). The hardness was determined by the Brinell method with an Innovatest hardness tester (Innovatest BV, Maastricht, The Netherlands); a tungsten carbide ball with a diameter of 2.5 mm and a load of 31.25 kg were used. The average hardness values and standard deviations were determined. Flexural strength tests were conducted using the three-point bending method on a Zwick Roell Z020 testing machine (Zwick AG, Ulm, Germany). The flexural test was carried out at room temperature and with a constant tool feed rate of 0.05 mm/s. The tests were conducted on three samples for each variant. Tests of the tribological properties were performed using a block-on-ring tribotester-T-05 (Figure 2) manufactured by ITEE (Radom, Poland). All the analyses were carried out in an air-conditioned room with constant humidity (about 30%) and a temperature of 21 °C. The T-05 tester allows materials to be tested in forward motion in dry and lubricated conditions. The lubricated sliding contact was conducted with TU-32 oil, dedicated for this type of alloys. In order to verify the behavior of the studied alloy under conditions simulating alternating starting and stopping of a turbine, “start-stop” tests were also performed on the sliding distance of 10,000 m. This test consisted of 10 consecutive friction cycles of 1000 m each using TU-32 oil sampled from the sump as the lubricant. TU-32 oil complies with DIN 51,515 part 1 and ISO 8068. The parameters of the tribological test are as follows:counter-sample—49.5 mm, steel 100Cr6, heat-treated to hardness of 55 HRCrotational speed—163 RPMload—50 Nsliding distance—1000 m, 10,000 m, 10 cycles × 1000 m.

A diagram of the tester is shown in Figure 2.

For each test variant, a minimum of three replications were performed. The applied tester enables the performance of tests in accordance with the methodology described in the ASTM standards: D 2714, D 3704, D 2981 and G 77. The tests were carried out in accordance with the guidelines contained in ASTM D 3704 [27], which mainly concern a counter-sample with a hardness recommended by the standard. The 50 N load is similar or almost identical to the loads encountered in real operating conditions of this type of bearing, while the increased load was used to determine the possibility of raising the operating parameters of this type of bearing employing the dedicated oil. Before weighing the samples after the test with TU-32 oil, it was removed from their surfaces. During the tribological test, friction force F was recorded continuously (which is necessary to determine the coefficient of friction) and the weight loss of the samples was determined based on the difference in weight before and after the tribological test. Additionally, the wear rate was determined. After the tests, the surfaces of the samples in the tribological contact area were analyzed and the observations were made using a scanning electron microscope.

## 3. Results and Discussion

### 3.1. Microstructure Characterization

Typical examples of the alloy microstructure in the initial state are shown in Figure 3 and Figure 4. This alloy is characterized by a multiphase microstructure with a uniform distribution of hard CuSn phase precipitates with the stoichiometric composition of Cu_6_Sn_5_, SnSb and fine lead particles, probably eutectic Sn-Pb, against the background of a matrix rich in tin. CuSn precipitates take the shape of both elongated needles and small irregular polygons. They are evenly distributed in the matrix, which is advantageous from the point of view of using this alloy mainly for bearing materials.

Detailed analysis of the chemical composition of individual precipitates and the matrix is presented below in the form of distribution maps of the alloying elements.

Micrographs of the microstructure of the SnSb9Cu4 (B89) alloy after FSP surface modification, made using a scanning electron microscope, are shown in Figure 5, Figure 6 and Figure 7. The presented micrographs taken at low magnification (Figure 5a and Figure 6a) show both the zone after pin passage and the area without modification, while the micrographs taken at greater magnification show the area after FSP modification (Figure 5b and Figure 6b). Even a cursory analysis of the micrographs of the microstructure indicates a change in the morphology of the precipitates present in the alloy, especially of the CuSn type. Their shape, size and arrangement have changed. It was found that the CuSn phase only locally retained an acicular form, but the vast majority after modification took the shape of small elongated grains with slightly rounded edges. Moreover, an increase in the rotational speed to 560 RPM favors a more even distribution of this CuSn phase. Refinement of the tin-rich matrix grains can also be observed. FSP causes an increase in temperature in the pin action impact area as a result of the frictional forces and leads to strong plastic deformation of this area. These phenomena probably also allow dynamic recrystallization to occur in the pin action impact zone.

The EDS analysis of the chemical composition (Figure 7) did not show any changes in the phase composition of the alloy as a result of friction modification, while the CuSn precipitate was significantly refined in the pin action impact area. The use of FSP modification also contributed to refinement of the eutectic Sn-Pb phases, which is a favorable phenomenon.

### 3.2. Mechanical Properties

The change in the dimensions, morphology and particle distribution of the CuSn, SnSb and Sn-Pb phases after FSP modification only slightly influenced the hardness of the studied alloy (Figure 8).

Modification of the B89 alloy by FSP resulted in a slight decrease (about 3 HB) in the hardness compared to the initial alloy (20 HB), which is probably related to alloy recrystallization occurring as a result of heating during frictional modification of its surface. There was no influence of the tool rotational speed on the hardness result. The microstructures of the studied alloy presented in Figure 5, Figure 6 and Figure 7 in the pin action impact area show a significant similarity in the morphology and distribution of the CuSn precipitates, which is also confirmed by the measurement of the hardness of the alloy.

The next stage of the research was the three-point flexural test of the studied alloy before and after FSP. The results are summarized in the form of a collective graph in Figure 9; additionally there are photos of the samples after bending.

The flexural strength values obtained as a result of the three-point flexural test (Figure 9) showed an increase in the case of the samples after FSP modification with a pin rotational speed of 280 RPM (307 MPa) compared to the alloy in the initial state, for which the flexural strength value was 214 MPa. The rise in flexural strength in the case of the B89 alloy after FSP modification with the speed of 280 RPM is caused by refinement of and a change in the morphology of the Cu_6_Sn_5_ phase particles to a more globular shape. Due to the significant pin action impact in the processed material and the dynamic flow of the material in its action impact zone, the stir zone is characterized by the presence of various stress states [28]. The direction of rotation and travel of the pin are the same on the advancing side, while on the retreating side they already show the opposite direction, which also generates an additional temperature gradient in the process zones. This causes the generation of additional stresses in the advancing zone in relation to the retreating side in the modified material [29,30]. Such a phenomenon significantly affects the susceptibility to cracking, as exemplified by the material modified with the pin rotational speed of 560 RPM, in which the strong pin action impact generated significant stresses, resulting in a significant reduction in flexural strength in the transition zone (123 MPa).

### 3.3. Tribological Properties

Figure 10, Figure 11, Figure 12, Figure 13 and Figure 14 below show the results of tests of the tribological properties of the B89 alloy, both in the initial state and after both FSP variants. Figure 10a,b present the curves showing the changes in the coefficient of friction as a function of the test time in lubricated friction conditions for the applied load of 50 N, while Figure 10b shows the course of changes in the coefficient of friction for the start-stop test, which is to simulate the actual operating conditions of a turbine (alternating starts and stops, or stops related to a system failure).

The analysis of the course of changes in the coefficient of friction under friction conditions with Tu-32 oil (Figure 10a,b) shows that up to approximately 500 s (sliding distance 250 m), the values of the coefficient of friction decrease significantly, which results from run-in and mutual alignment of the surfaces. This is the time period in which mutual run-in of the sample surface and possibly counter-samples in the tribological contact surface takes place, and the value of the coefficient of friction is a function of the surface morphology, which depends on the size and type of individual precipitates present on the studied surface. After run-in, the coefficient of friction stabilizes and the curves show slight changes in their course. On the sliding distance of 1000 m (Figure 10a), after an initial rapid decrease in the friction coefficient over the entire range, its value decreases slightly. This waveform is consistent with that observed on the 10,000 m sliding distance (Figure 10b), where after about 1000 m there is stabilization or even a slight rise in the coefficient of friction (material after FSP 280 RPM). This corresponds to the second range of changes in the frictional force on the Stribeck curve, in which the oil is spread over the friction surfaces and the formation of a continuous insulating layer. The coefficient of friction drops below 0.1, which confirms the correct selection of lubricant. The lowest values, regardless of the sliding distance, are the coefficient of friction for the material after passage of the pin at 560 RPM, while the curves for the initial material and the material after passage of the pin at 280 RPM have a similar character and a similar course. Such a stable course of the curves proves the correct operation of the oil film, which properly insulates the rubbing surfaces, preventing the occurrence of adhesive wear. Minor disturbances in the course of the curves may indicate the crumbling of particles formed as a result of refinement of CuSn acicular particles. The course of changes in the coefficient of friction for the start-stop test can be characterized in a similar way. For each 1000-m cycle, the factor decreases after start-up. At the moment of stopping and restarting, the value of the coefficient remains in the range of the lubricated friction value; thus, oil remains in the friction pair. Nevertheless, its amount at the time of restarting is not sufficient to completely isolate the rubbing surfaces. The greatest decrease in the coefficient is observed in the entire first cycle; in subsequent cycles the course is stable (initial material and after FSP 560 RPM) or increasing, similar to the continuous course for the material after FSP with a pin speed of 280 RPM. This increase in the coefficient of friction may be the result of an insufficient degree of recrystallization of the matrix and the attachment of new precipitates formed as a result of crushing the needles in it, which causes them to crumble and the matrix to deform to a greater extent than the highly recrystallized material in the case of using the higher rotational speed of the pin.

Figure 11 shows the average coefficients of friction for individual material states and for various test parameters in the presence of oil.

The analysis of the average values of the coefficient of friction presented in Figure 11 confirms that the application of FSP modification reduces its value, regardless of the applied parameters of the tribological test, especially in conditions of intermittent friction, which can be extremely advantageous in friction pairs operating intermittently and the possible design problem associated with the continuous supply of oil to the friction pair. In the case of the B89 alloy, there is a stronger effect of FSP than in the case of the B83 alloy, which was demonstrated in the studies presented in publication [26].

When analyzing the surface after friction presented in Figure 12, Figure 13 and Figure 14, it can be concluded that the dominant wear mechanism in the studied alloy is abrasive wear; in the initial material clear scratches running parallel to the direction of friction with a slight degree of depression in the matrix are visible. They are the result of the impact of the present 100Cr6 steels of hard Cr_23_C_6_ type carbides on the B89 surface. On the surface, one can also observe bright precipitates that differ significantly from the matrix and the precipitates present in it. In order to determine the type of these precipitates, maps of the distribution of elements on the friction surface were made (Figure 15). The presented analysis of the chemical composition on the surface showed the presence of iron oxides and tin oxides. According to the Ellingham–Richardson diagram, they are the most probable oxides in the investigated alloy. The presence of iron on the surface is a result of its diffusion from the pin into the tin matrix. These oxides present on the surface participating in friction do not form a strong bond with the substrate and easily detach from the surface, and can therefore participate in friction. When analyzing the surface around the oxides, one can see traces of their movement in the form of grooves, but their size is definitely smaller than those resulting from the influence of carbides from the counter-sample. This proves their insignificant influence on the course of friction, which is also related to their low mechanical properties.

Comparing the morphology of the sample surfaces after FSP modification to the initial material, it can be concluded that the change in the morphology of the precipitates resulting from the impact of the rotating pin affects the course of friction and the type of wear mechanisms. In the matrix area, the dominant mechanism is frictional wear; similar to the initial material, cracks parallel to the course of friction can be seen. However, there are also clear traces of chipping of the CuSn phase particles, which, as shown in the presented photos (Figure 13 and Figure 14), after the modification process, are not so strongly fixed in the matrix or interconnected with other particles formed during pin action impact. The mutual integrity and with the matrix depends on the rotational speed of the pin; the use of 560 RPM is associated with a greater plastic action impact on the material being processed and also the thermal action impact in this zone. The fraction of chippings in the material processed with the pin speed of 560 RPM (Figure 14) is much lower than in the material processed with the speed of 280 RPM; they are also smaller in terms of their surface and depth, which proves their better cohesion with the matrix. In Figure 13a, a groove in the direction of friction can be seen, the size of which is larger than the scratches resulting from abrasive wear, possibly indicating the scuffing mechanism that begins in the observed area, characteristic for this type of systems. The refinement of the CuSn phase precipitates and recrystallization of the matrix also affect the lack of oxide particles on the friction surface, so clearly visible on the surface of the initial material (Figure 12). It also results in virtually no smearing of the tin-rich and soft matrix on the friction surface, which is traceably present in the initial material—especially in large areas without precipitates.

In order to compare the tribological properties of the studied materials, the weight loss during the test (Figure 16) and the wear rate (Figure 17) were also determined. 

Considering the abrasion resistance from the point of view of weight loss, it can be concluded that both the 280 RPM and 560 RPM pin speed modification cause smaller weight losses during the tribological test. These losses are insignificant and it is also the result of proper selection of the lubricant in the form of TU-32 oil, which, according to the authors’ own research, reduces the weight loss by over 140 times compared to dry friction on a 1000 m distance. The smallest weight losses, regardless of the friction conditions, were obtained by the samples made of the alloy after friction modification with the pin rotational speed of 560 RPM, which is a result consistent with the recorded course of changes in the coefficient of friction. Strong refinement and better fixation in the matrix of new smaller and partially rounded particles contribute to increasing the wear resistance in the modification area. The results are also confirmed by the determined wear rate and the lowest wear rate, and despite the chippings present (Figure 14 and Figure 17), it was recorded for the alloy samples after friction modification with the pin rotational speed of 560 RPM. The material is used up the fastest in the initial period, in the presented tests during the first 1000 m; after exceeding this value, the friction stabilizes and both the weight loss and the wear rate are much lower than in the initial stage.

## 4. Conclusions

Based on the studies of the microstructure, selected mechanical properties and tribological tests, the following conclusions were drawn:The application of FSP modification to the B89 alloy affects refinement of and a change in the morphology of CuSn precipitates, and the best results were achieved with the pin rotational speed of 560 RPM.The modification reduces the hardness of the studied alloys, which is the result of recrystallization of the matrix during the operation of the pin, while the flexural strength increases; nonetheless, in the case of the alloy modified with the pin rotational speed of 560 RPM, cracking is located in the transition zone, limiting the beneficial effect of recrystallization and refinement of the particles as a result of this process.In the case of the B89 alloy under lubricated friction conditions, the use of FSP improves the coefficient of friction in relation to the starting material most strongly in the case of the alloy after modification with the pin rotational speed of 560 RPM.The weight loss, which is one of the measures of wear resistance, decreases as a result of FSP; the higher the pin rotational speed used, the more the weight loss decreases, i.e., it leads to improvement of the wear resistance of the studied alloys under specific tribological test conditions.

## Figures and Tables

**Figure 1 materials-14-02627-f001:**
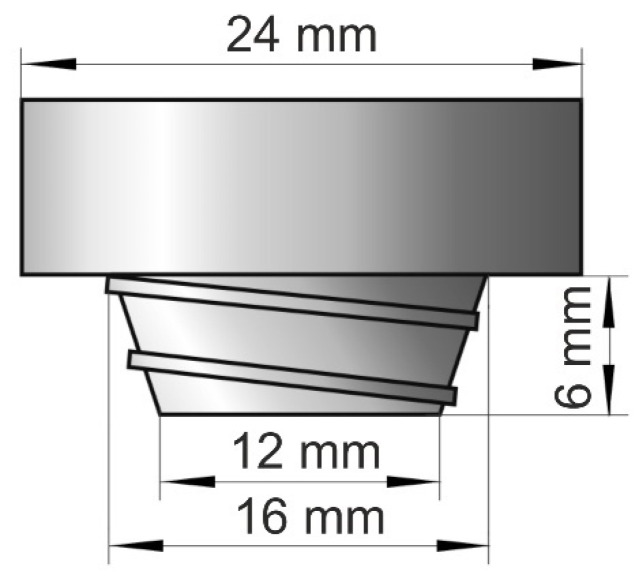
Diagram of Whorl pin used to modify surface of B89 alloy.

**Figure 2 materials-14-02627-f002:**
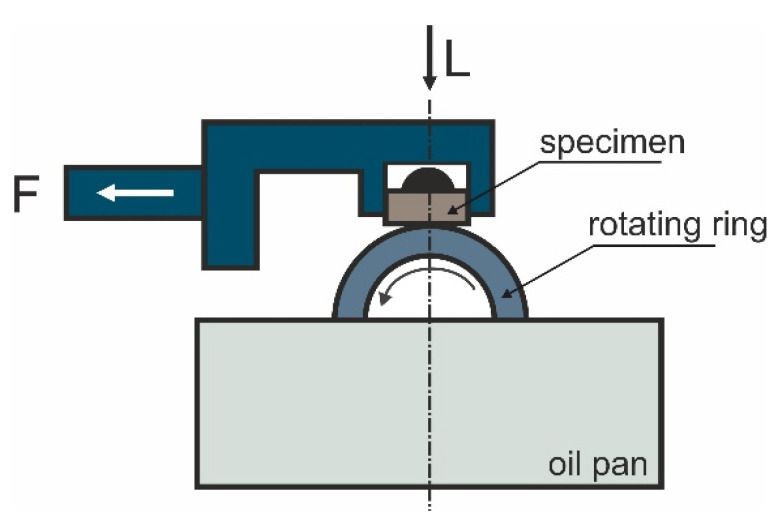
Diagram of test using T-05 tribometer.

**Figure 3 materials-14-02627-f003:**
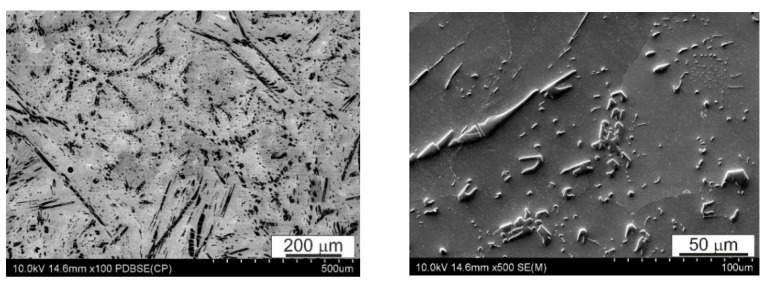
Microstructure of B89 alloy in its initial state, SEM.

**Figure 4 materials-14-02627-f004:**
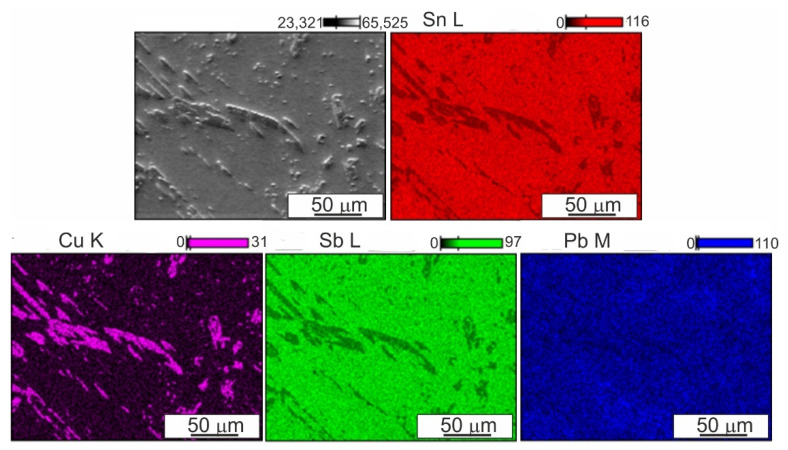
Microstructure and distribution maps of alloying elements (Sn, Cu, Sb, Pb) in B89 alloy in initial state, SEM EDS.

**Figure 5 materials-14-02627-f005:**
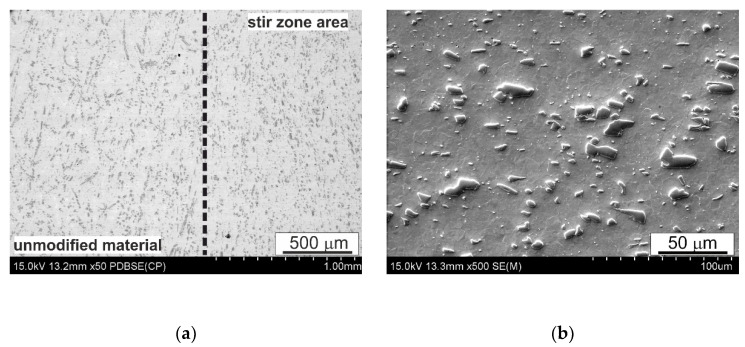
Microstructure of SnSb9Cu4 alloy after FSP; 280 RPM; (**a**) figure show both the zone after pin passage and the area without modification, (**b**) figure show stir zone area.

**Figure 6 materials-14-02627-f006:**
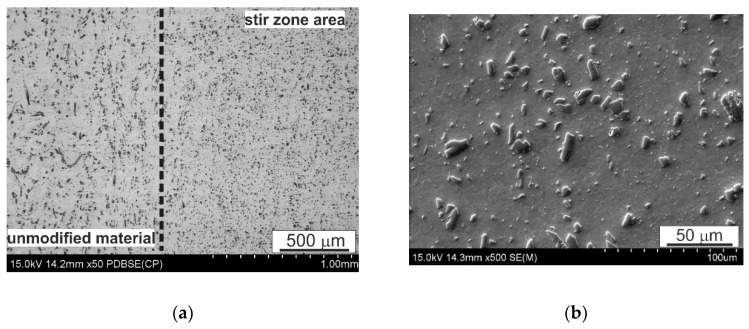
Microstructure of SnSb9Cu4 alloy after FSP; 560 RPM; (**a**) figure show both the zone after pin passage and the area without modification, (**b**) figure show stir zone area.

**Figure 7 materials-14-02627-f007:**
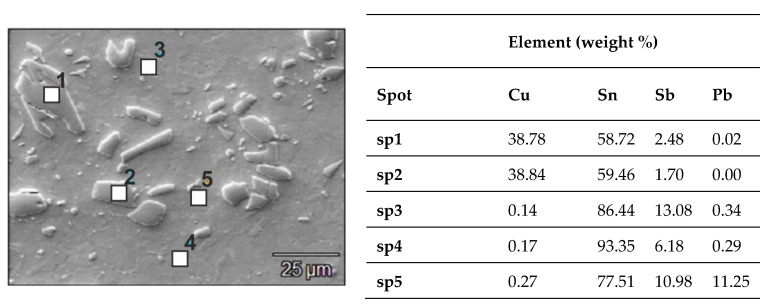
Microstructure of B89 alloy after FSP modification at 280 RPM and results of EDS chemical composition spot analysis; SEM.

**Figure 8 materials-14-02627-f008:**
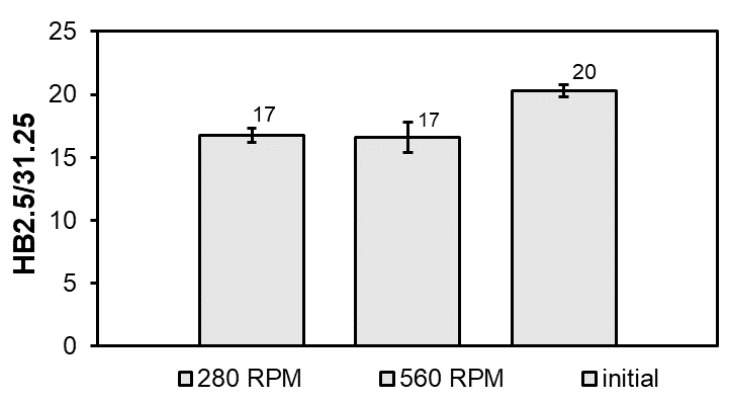
Comparison of hardness of B89 alloy before and after FSP.

**Figure 9 materials-14-02627-f009:**
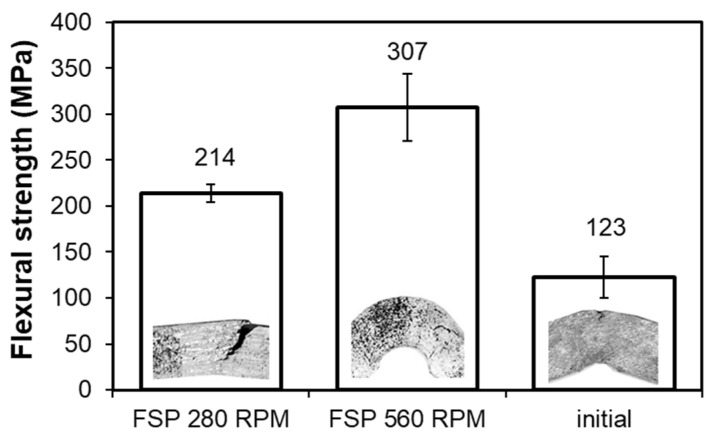
Comparison of flexural strength and fracture morphology of B89 alloy before and after FSP.

**Figure 10 materials-14-02627-f010:**
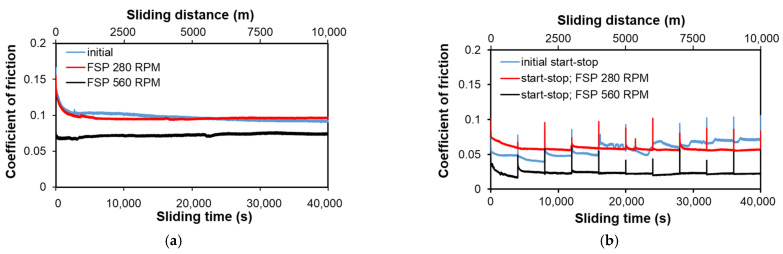
Coefficient of friction of B89 alloy before and after FSP as a function of time and processing conditions; (**a**) lubricated friction, load 50 N, sliding distance 10,000 m; (**b**) lubricated friction, load 50 N, start–stop 10 × 1000 m.

**Figure 11 materials-14-02627-f011:**
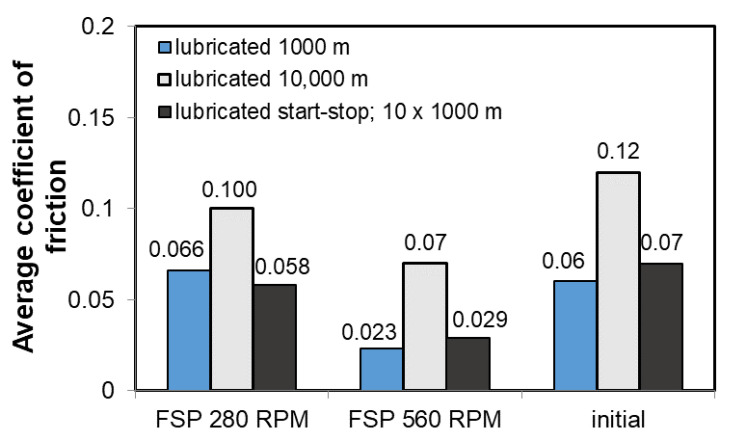
Coefficient of friction determined at load of 50 N.

**Figure 12 materials-14-02627-f012:**
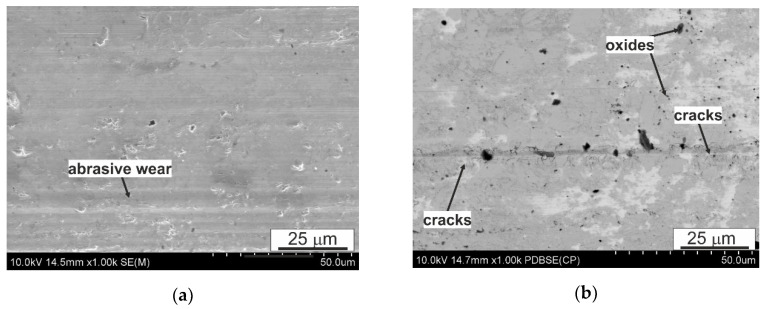
Characteristic sample surface of B89 alloy after tribological tests in lubricated friction, (**a**) 10,000 m; (**b**) start-stop 10 × 1000 m, SEM.

**Figure 13 materials-14-02627-f013:**
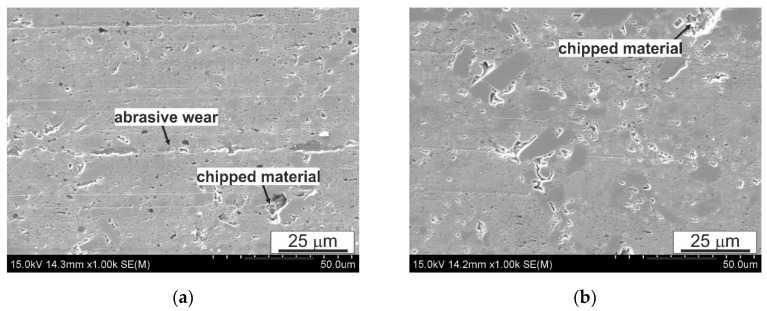
Characteristic sample surface of B89 alloy (FSP 280 RPM) after tribological tests in lubricated friction, (**a**) 10,000 m, (**b**) start-stop 10 × 1000 m; SEM.

**Figure 14 materials-14-02627-f014:**
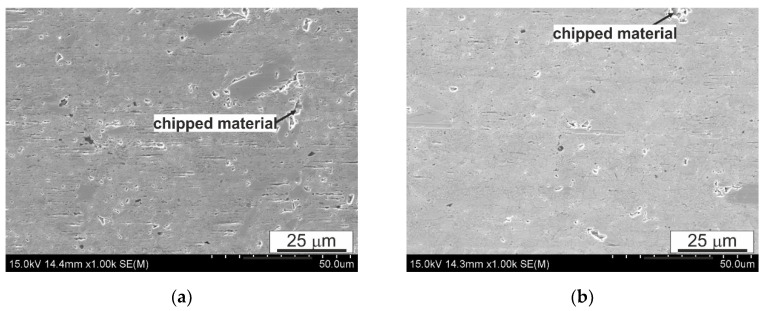
Characteristic sample surface of B89 alloy (FSP 560 RPM) after tribological tests in lubricated friction, (**a**) 10,000 m, (**b**) start-stop 10 × 1000 m; SEM.

**Figure 15 materials-14-02627-f015:**
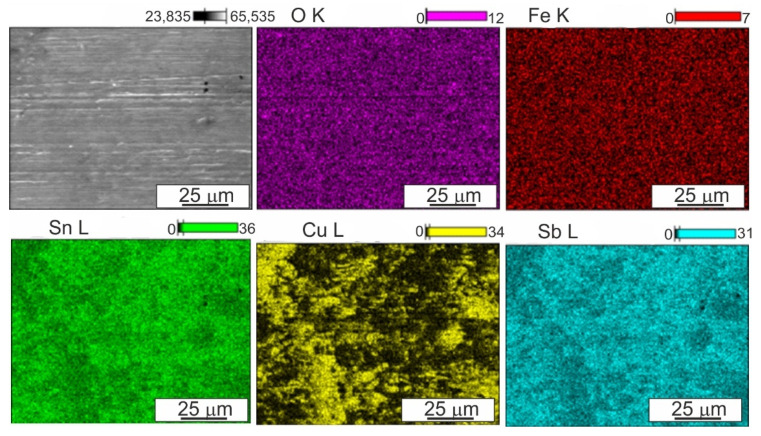
Surface after friction of B89 alloy in initial state and maps of distribution of elements: O, Fe, Sn, Cu, Sb; lubricated friction; SEM EDS.

**Figure 16 materials-14-02627-f016:**
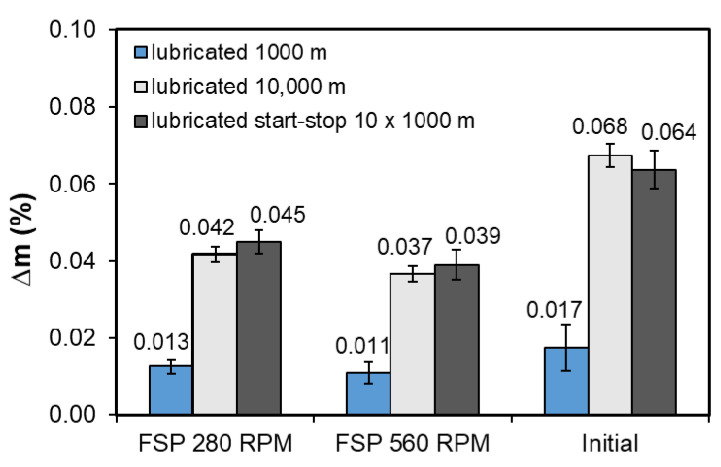
Weight loss determined at load of 50 N.

**Figure 17 materials-14-02627-f017:**
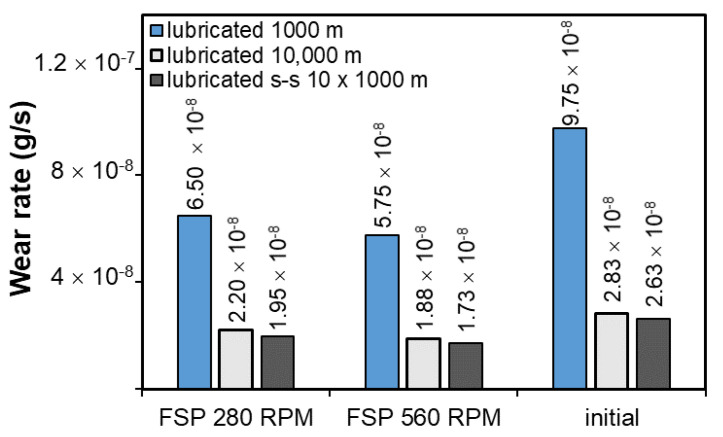
Wear rate of B89 alloy as a function of time and process conditions.

**Table 1 materials-14-02627-t001:** Chemical composition of B89 alloy.

Element	Weight (%)	Atom (%)
Cu	6.20	11.02
Sn	85.43	81.38
Sb	7.90	7.33
Pb	0.48	0.26
Total	100.00	100.00

## Data Availability

Data is contained within the article.

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
