# Peer review of "Effect of FSP on Tribological Properties of Grade B89 Tin Babbitt"

_materials, 2021, doi:10.3390/ma14102627_

Round 1
Reviewer 1 Report
Dear Authors,
Your paper presents an interesting study on a B89 tin babbitt material with surface modification by FSP using a high-speed stell whorl pin.
Please take into consideration de following comments :
Major corrections :
- To enhance the reading experience, I suggest that you split your section "Materials and methods" in two independant sections "Methods" and "Materials".
- The miscrostructure of the samples before the tribological tests should be presented before the results section (e.g. in the material section).
- More generally, if the aim of your study is the tribological characterization of your samples, it would be better for the reading that the other tests (mechanical tests or miscrostructure characterisations before tribological testing) were presented befor the result section.
- As for the figure8 and 9, the other bar graphs (fig. 11, 16 and 17 should present error bars to higligh that the differences observed in your results are significant.
Minor corrections :
- Please performe a small english spell check on the paper. For instance , at line 29, "refinement of and" (extra "of")
Sincerely,
Author Response
Dear Reviewer,
We greatly appreciate your thoughtful remarks that helped improve the manuscript. In the following, we give a point-by-point reply to your remarks. The changes and amendments have been introduced to the text of the publication. All the changes, I have highlighted by using the „Track Changes”. The article is also revised by a native speaker.

Reviewer 2 Report
The manuscript is in overall clear, the methodology used was properly designed and well performed. Nevertheless, there are different unclearness and inconsistencies that must be addressed. See the attached pdf with comments

Author Response

(The authors gave the same response as above.)
